# In-flight emission measurements with an autonomous payload behind a turboprop aircraft

Gregor Neumann<sup>1,2</sup>, Andreas Marsing<sup>1</sup>, Theresa Harlass<sup>1</sup>, Daniel Sauer<sup>1</sup>, Simon Braun<sup>1</sup>, Magdalena Pühl<sup>1</sup>, Christopher Heckl<sup>1</sup>, Paul Stock<sup>1</sup>, Elena De La Torre Castro<sup>1,5</sup>, Valerian Hahn<sup>1</sup>, Anke Roiger<sup>1</sup>, Christiane Voigt<sup>1,2</sup>, Simon Unterstraßer<sup>1</sup>, Jean Cammas<sup>3</sup>, Charles Renard<sup>3</sup>, Roberta Vasenden<sup>4</sup>, Arnold Vasenden<sup>4</sup>, and Tina Jurkat-Witschas<sup>1</sup>

**Correspondence:** Gregor Neumann (gregor.neumann@dlr.de)

## Abstract.

This paper reports on the successful first deployment of a new, autonomously operating measurement system on a Grob G 520 Egrett aircraft, which was used as a chase aircraft to perform in-flight aerosol and trace gas measurements of engine exhaust from other aircraft. A suite of custom-built and commercially available instruments was selected, modified, and adapted to operate in the unpressurized compartment of the Egrett over a wide range of ambient temperatures and pressures. We performed these first in-flight emission measurements at cruise altitudes between 7.6 and 10.4 km (FL250 and FL340) behind a Piper Cheyenne, a twin-turboprop aircraft powered by Garrett/Honeywell TPE 331-14 engines over Texas in April 2022. The instrumentation and inlets on the Egrett were designed to measure non-volatile particulate matter ( $tPM_{D_p>10}$ ), nitrogen oxides (NO and  $trace{NO}_2$ ), water vapor ( $trace{H_2O}_2$ ), carbon dioxide ( $trace{CO}_2$ ), and contrail ice particles. All instruments were operated in relevant plume conditions at cruise altitudes and distances ranging from 100 to 1200 m between the two aircraft. The instruments proved to have high reliability, a large dynamic range, and sufficient accuracy for measuring the emissions of the turboprop engine.

We derived the emission indices (EI) for tPM, nvPM, and  $NO_x$  at cruise. The particulate emission indices range from 9.6 to  $16.2 \times 10^{14} \, \mathrm{kg^{-1}}$  (particles per kg fuel burned) for  $\mathrm{EI_{tPM}}$  and from 8.1 to  $12.4 \times 10^{14} \, \mathrm{kg^{-1}}$  for  $\mathrm{EI_{nvPM}}$  (medians). For  $NO_x$  we find rather low  $\mathrm{EI_{NO_x}}$  between 7.3 and 7.7 g kg<sup>-1</sup> for  $\mathrm{EI_{NO_x}}$  (medians). Furthermore, the tPM aerosol size distributions have been measured in the exhaust plume, taking into account the size-resolved sampling efficiency of the instrument. The analysis of the size-resolved emission index indicates a log-normal distribution with geometric mean and standard deviation at  $D_g = 27.5 \pm 2.0 \, \mathrm{nm}$ . This geometric diameter value is in the range of jet engine soot emissions previously measured in flight. The measurements help to constrain the climate impact of small-class turboprop engines and need to be compared to larger turboprop aircraft in the future. The current work provides a benchmark for future alternative  $H_2$  propulsion systems, such as fuel cells and direct combustion engines.

<sup>&</sup>lt;sup>1</sup>Deutsches Zentrum für Luft- und Raumfahrt, Institut für Physik der Atmosphäre, Oberpfaffenhofen, Germany

<sup>&</sup>lt;sup>2</sup>Institute of Atmospheric Physics, Johannes Gutenberg-Universität, Mainz, Germany

<sup>&</sup>lt;sup>3</sup>Airbus Operations SAS, Toulouse, France

<sup>&</sup>lt;sup>4</sup>AV Experts LLC, Texas, US

<sup>&</sup>lt;sup>5</sup>Faculty of Aerospace Engineering, Delft University of Technology, Delft, the Netherlands

## 1 Introduction

Assessing the climate impact of aviation requires knowledge of emissions and contrails from current technologies, including the regional sectors. Future aircraft powered by hydrogen-based propulsion systems, including fuel cells and direct hydrogen combustion engines, could eventually replace short-haul fossil fuel-based turboprop aircraft in the long term. However, significant uncertainties remain regarding the climate impact of the current regional fleet under cruising conditions, due to the lack of in-flight measurements and a public emissions database. Aviation accounts for approximately 3.5% of total anthropogenic effective radiative forcing (Lee et al., 2021). Of this, about one-third results from  $CO_2$  emissions (34 mW m<sup>-2</sup>) accumulated since the beginning of modern aviation, while the remaining two-thirds result from non- $CO_2$  effects, like  $NO_x$  emissions (17 mW m<sup>-2</sup>) and contrail cirrus formation (57 mW m<sup>-2</sup>) (Lee et al., 2021). According to the European Aviation Environmental Report 2022 (EASA, 2022), in 2019 75% of all flights from European airports were in the medium and short range below 1500 km, and 9.8% of all flights were turboprop engine aircraft.

Both turboprop and turbofan engines are based on the gas turbine principle, and the combustion processes are similar (Bräunling, 2015). However, the mixing and dilution in the wake of the aircraft are expected to differ from jet engine exhaust due to the effect of the propeller and the expulsion of the emissions. This, in particular, may affect contrail properties like the initial ice crystal number.

Due to their higher fuel efficiency and lower operating costs, turboprops can still compete in the short to medium-range sector with the turbofan engines that dominate global aviation. Turboprop engines are lighter, simpler in operation, generate high power per unit weight, and have better take-off and landing performance than turbofan and turbojet engines (FAA, 2024). Turboprop aircraft are the most efficient at lower speeds (between approx. 400 and 650 km/h) and lower altitudes (between approx. 5500 and 9100 m). Therefore, due to the low emission altitude of these aircraft types, their impact on non-CO<sub>2</sub> effects is expected to be less relevant to the overall radiative forcing from aviation. Keles et al. (2024) argue that turboprops are able to reduce the CO<sub>2</sub> and non-CO<sub>2</sub> effects at short ranges of ~740 km compared to single-aisle turbofan aircraft, despite having a much lower payload. Maruhashi et al. (2024) shows that the NO<sub>x</sub> effects on the radiation forcing mainly depend on the altitude of emission. Future aircraft similar in size, power, and altitude range to turboprop aircraft may be the first to be equipped with new disruptive technologies such as hydrogen fuel cell electric propulsion systems (International Air Transport Association, 2023; Federal Aviation Administration, 2023). The contrail formation altitude depends on ambient conditions, engine efficiency, and engine technology. Hydrogen combustion and fuel cell propulsion enable contrail formation at higher ambient temperatures than kerosene combustion (Schulte and Schlager, 1996; Bier et al., 2024) according to the Schmidt-Appleman criterion. While for classical kerosene combustion, contrail formation is limited by thermodynamic constraints, H<sub>2</sub> contrail formation is limited by droplet freezing. In aircraft plumes, liquid droplets freeze in the temperature range of 230 K to 235 K, depending on the droplet properties (Zink et al., 2025; Bier et al., 2024). For higher ambient temperatures, contrails remain liquid, evaporate quickly after droplet formation, and will not be persistent (Gierens, 2021; Kaufmann et al., 2024). A benchmark against current technologies is therefore needed to assess the potential benefits of these new aircraft in terms of  $CO_2$  and non- $CO_2$  effects.

The ICAO aircraft engine emission database includes turbojet and turbofan engine types for a static thrust greater than 26.7 kN for which emissions are regulated (ICAO, 2023). As a consequence, little public information on turboprop emissions exists. To a large extent, the emission data are proprietary to engine manufacturers and operators, making it difficult to incorporate them into a global aviation climate assessment. Also, turboprop emission data are, if at all, mostly available for sea level pressure conditions. Due to a lack of in-flight emission measurements, the scalability of ground emissions to emissions at altitude using fuel flow methods has not been investigated. It is also unclear to what extent ground-based emission data are related to engine data at altitude (Döpelheuer and Lecht, 1998; Schulte and Schlager, 1996; Dischl et al., 2024; Märkl et al., 2024; Harlass et al., 2024).

Airborne measurements of aircraft emissions during cruise are costly and challenging and therefore only a limited number of these measurements are reported in the literature, e.g. Fahey et al. (1995); Schulte and Schlager (1996); Schlager et al. (1997); Schumann (2000); Voigt et al. (2010); Jurkat et al. (2011); Voigt et al. (2012). Recent measurements have mainly been reported for jet engine aircraft using the DLR Falcon or the NASA DC8 as chase aircraft (Moore et al., 2017; Bräuer et al., 2021a, b; Voigt et al., 2021; Dischl et al., 2024; Harlass et al., 2024; Märkl et al., 2024).

The adaptation of measurement instruments for deployment on research aircraft presents several challenges. They must be compact and lightweight to accommodate space and weight constraints while also meeting strict safety requirements. In the specific case of measuring aircraft emissions during formation flights, instrument requirements are defined by their robustness to withstand highly turbulent conditions and to operate at extreme temperatures below -40 °C, and pressures below 500 hPa. As the market for these instruments is limited, adapting ground-based measurements to altitude often requires specific modifications. Furthermore, they require high temporal resolution, accuracy, and a broad dynamic range to capture rapid fluctuations in emissions. The variability of atmospheric aerosol and trace gas background concentrations is often within 1 to 3 orders of magnitude, depending on the species measured (Kaufmann et al., 2016; Brock et al., 2021; Dischl et al., 2022; Voigt et al., 2022; Tomsche et al., 2022; Harlass et al., 2024; Jurkat-Witschas et al., 2025). However, aircraft exhaust plumes in the near-field contain aerosol concentrations several orders of magnitude higher, depending on the dilution of the emitted plume (Kärcher et al., 1996; Kärcher and Yu, 2009). Since the speed envelope of the emitting aircraft must match that of the chasing aircraft, suitable aircraft pairings are necessary. In particular, smaller turboprop aircraft often fall outside the speed range of turbofan-powered aircraft, limiting viable combinations.

To provide a broad picture of non- $CO_2$  effects from aircraft emissions, various parameters such as water vapor,  $CO_2$ , and  $NO_x$  mixing ratio, as well as nvPM number concentration, tPM number concentration and size distribution, and ice particle measurements form the basic components of an in-flight payload on a chaser aircraft. We report here on a comprehensive set of autonomous in situ instruments for contrail and emission measurements aboard the Grob Egrett. We provide measurements of  $CO_2$ ,  $H_2O$ ,  $NO_x$ , tPM, nvPM in the wake of a Cheyenne Piper turboprop aircraft. To the best of our knowledge, these are the first quantitative emissions measurements behind a turboprop aircraft in flight. The DLR payload aboard the Egrett presented in this paper will be the basis for the Blue Condor (German Aerospace Center (DLR), 2022; Airbus, 2022) measurements, investigating the contrail properties of a small hydrogen turbojet aircraft.

# 90 1.1 Campaign overview






The emission data were obtained as part of a flight test campaign conducted by Airbus from April 4, 2022, to April 14, 2022, based in Denison, Texas, USA. The chase aircraft, a Grob G 520 Egrett (Fig. 1), is a high-altitude and long-endurance turboprop aircraft with a certified maximum operating altitude of 15,240 m (50,000 ft) (Grob Aircraft SE), a maximum airspeed of 463 km/h (250 kn), and a range of 4260 km (2,300 Nmi) with an endurance of 8.0 hours dependent on payload and weather (NASA Airborne Science Program). Operated by AV Experts LLC, the Egrett was suited to test the instruments and to perform measurements in the near-field exhaust plume (100 - 1200 m) and background atmosphere. The instruments for contrail and emission measurements were installed and operated by the German Aerospace Center (DLR). As the Egrett is a single-pilot aircraft, the instruments were started shortly before the flight and worked autonomously without further interaction from the pilot or the operators. The campaign comprised 6 flights in 12 days. In addition to test and chase flights with other emission aircraft, we were able to conduct two near-field emission flights behind a Garrett/Honeywell TPE 331-14 twin-engine (each rated at 1213 kW maximum power) turboprop aircraft of type Piper Cheyenne 400LS (Fig. 2), also operated by AV Experts LLC. In the following section, the instruments for aerosol (tPM, nvPM, and size distribution), NO and  $NO_2$  ( $NO_x$ ),  $H_2O_x$ , and  $CO_2$  measurements are described in detail. All instruments are installed in the fuselage of the Egrett in an unpressurized compartment. The belly of the Egrett was extended to accommodate the  $NO_x$ -SIOUX instruments (Fig. 1).

A 2.5 m long mast positioned on the upper fuselage of the Egrett held a forward-facing isokinetic aerosol inlet and two backward-facing inlets for CO<sub>2</sub> and H<sub>2</sub>O. These inlets were connected to the instruments in the belly of the aircraft with heated stainless steel tubing. The inlet position was optimized to avoid the influence of the propeller and the emissions of the chase aircraft during sampling. A second sampling position was selected in front of the left landing gear at the left wing. The Cloud-, Aerosol-, and Precipitation Spectrometer (CAPS) was integrated in a canister next to two sampling lines for NO<sub>x</sub> and H<sub>2</sub>O. The influence of the Egrett's propeller on the measurement was visualized by placing tufts along the mast and the CAPS. During ground tests and in flight, they were monitored to see if and how far the propeller wash affected the air stream. As the tufts at the inlets did not move during ground test and with pitch and roll during the flight, we estimate the effect of the propeller at the measurement locations to be negligible. The configuration of two inlet positions (in front of the landing gear and on top of the mast) is part of the Blue Condor Project (German Aerospace Center (DLR), 2022; Airbus, 2022). These inlets are designed to measure contrail ice crystals, NO<sub>x</sub>, and H<sub>2</sub>O emissions from a H<sub>2</sub>-combustion engine. Simultaneously, they determine the background aerosol concentration and size distribution to assess the dependence of background aerosol on contrail properties (Kärcher, 2018; Bier et al., 2024). Additionally, in-plume measurements from the mast inlet position allow us to probe kerosene engine emissions like soot and CO<sub>2</sub> to derive emission indices. As the atmospheric conditions for conventional contrail formation were not met during the campaign, the contrail ice particle measurements from the CAPS wing probe (Kleine et al., 2018; Bräuer et al., 2021c; Märkl et al., 2024) are not discussed in this work.

Figure 1. The chaser aircraft Grob G 520 Egrett equipped with instruments for contrail and emission measurements. The aircraft was modified with a mast holding the inlet for aerosol,  $CO_2$ , and water vapor measurements connected to instruments inside the fuselage. The main compartments in the fuselage contain the A-Box, the WARAN, and the CR2 water vapor instrument. The SIOUX instrument for  $NO_x$  measurements is housed in the belly of the Egrett. The left landing gear holds the CAPS instrument for ice crystal detection as well as sampling lines for  $NO_x$  and water vapor leading to the SIOUX and WARAN instruments, respectively.

**Figure 2.** A picture taken from the Egrett's left landing gear camera, with the propeller of the Egrett (on the right) and the CAPS probe (on the top), shortly before going into formation flight with the Piper Cheyenne (400LS, registration 30 N92EV). The emission aircraft equipped with a two-engine turboprop of type Garrett/Honeywell TPE 331-14 was chased at altitudes between 7.6 and 10.4 km.

## 2 Instrumentation





The Egrett was equipped with instruments for the measurement of aerosol (tPM, nvPM, and size distribution), NO and  $NO_2$  ( $NO_x$ ),  $H_2O$ , and  $CO_2$ . In this section, we describe the different measurement principles, the modifications made to adapt the instruments to the Egrett, the characterization and calibration of the instruments, and their specific uncertainties. Figure 1 provides an overview of the location of the instruments on the aircraft.

## 2.1 Aerosol measurements

#### 2.1.1 Aerosol-Box (A-Box)

The Aerosol-Box (Fig. 3), further referred to as A-Box, is a custom-made sealed aluminum box of size  $0.54\,\mathrm{m}$  x  $0.85\,\mathrm{m}$  holding all aerosol and  $\mathrm{CO}_2$  instrumentation in the unpressurized compartment of the Egrett. Aerosol measurements include tPM and nvPM number concentrations as well as total particle size distribution measurements. A pressure-stabilized environment was required for all instruments to ensure stable sampling conditions. Hence, the A-Box was tested for its compressive strength and equipped with a manometric switch set to ground-level pressure combined with a high pressure  $\mathrm{N}_2$  bottle to compensate for any small leaks. During the flight, the pressure inside the A-Box varied between 970 and 1020 hPa. Ambient temperatures ranged from -28 to -48 °C, while the internal temperature of the A-Box increased from 15 to 35 °C due to the heat emitted by the instrumentation.

Figure 3 shows the flow plan of the A-Box. The A-Box contains three advanced Mixing Condensation Particle Counters (aMCPC), Brechtel Model 9403. The first aMCPC is used to determine the tPM concentration and the second, with an upstream thermodenuder consisting of a heated section followed by a cooled section to evaporate the volatile aerosol, to determine the nvPM concentration. A third aMCPC was used in combination with a miniature Scanning Electrical Mobility Sizer (mSEMS, Brechtel Model 9404, Fig. 4) to measure aerosol size distributions in a range from 5 to 350 nm.

An Optical Particle Counter (OPC, Grimm SkyOPC model 1.129) was also installed in the A-Box to measure the size distribution of larger aerosols in the range from 250 to 3000 nm.

For detecting  $CO_2$  mixing ratios, a high frequency (5 Hz) non-dispersive infrared gas analyzer (Licor-7000, LI-COR (b)) is included in the A-Box.

## 145 2.1.2 Advanced Mixing Condensation Particle Counter (aMCPC)

The advanced Mixing Condensation Particle Counter (aMCPC), Model 9403 from Brechtel Manufacturing Inc., described in Mei and Goldberger (2020) detects aerosol particles over a large size range. Due to their light weight (1.8 kg), small size (0.18 m x 0.12 m x 0.13 m), low power consumption (avg. 9 W), and independent operation, the aMCPC was selected for the Egrett adaptation. It requires 10 - 14 V DC, supplied by the aircraft. At 1 atm, the lower 50% detection efficiency is reached at a diameter of 7 nm. As particles smaller than 50 nm are difficult to detect optically, the aMCPC uses a chamber with a supersaturated vapor of high-purity n-butanol to grow particles by heterogeneous nucleation to a detectable size of several microns (Ahn

Figure 3. A-Box containing aerosol, and  $CO_2$  instruments and its flow chart. In flight, the box is sealed with side panels and a pressure gauge controls a switch connected to a pressurized  $N_2$  bottle to maintain constant pressure in case of minor leakage.

and Liu, 1990; Stolzenburg and McMurry, 1991). Unlike conventional laminar flow CPCs, the mixing condensation particle counter uses turbulent mixing of the so-called sample flow with the saturator flow, (Kousaka et al., 1982). This results in a fast response time of 180 ms, which is useful for our in-flight measurements. The saturator flow is a clean, filtered airflow that passes through the heated saturator chamber (47 - 57 °C) where it becomes saturated with butanol vapor. The sample airflow is mixed in the cylindrical condenser chamber (21.9 - 31.9 °C), where the butanol vapor supersaturates and condenses on the sample air's particles. The new combined flow passes through the optics block, and the grown particles are detected by light scattering from an infrared laser. A critical orifice at the exit and a vacuum pump downstream ensure a constant flow of 0.72 lpm through the instrument. Assuming that the measurements are conducted under low-pressure conditions of 400 hPa or less at the instrument's inlet, as is the case during in-flight emission measurements, it is essential that the same pressure is maintained in both the sample and saturator flow lines. Hence, a bypass separates the saturator flow from the sample inlet line.


Figure 4. a) ① Two aMCPC for tPM and nvPM number concentration. ② mSEMS in combination with an aMCPC and X-ray charge neutralizer for tPM size distribution measurements. ③ X-ray source. ④ Thermodenuder to evaporate volatile aerosol upstream of a aMCPC. ⑤ Licor for  $CO_2$  measurements. b) WARAN and CR2 for water vapor measurements. c) mSEMS (right) in combination with an aMCPC (left). d) CAPS probe for ice particle measurements and  $H_2O/NO_x$  inlet at the landing gear. e) Thermodenuder. f) StratospherIc Observation Unit for nitrogen oXides (SIOUX) used for NO and  $NO_2$  measurements. g) Top of the mast inlet. (Forward-facing aerosol inlet and two backward-facing inlets for water vapor and  $CO_2$ )

A filter then purifies the flow before it enters the saturator chamber. Laminar flow elements record the sample and saturator flow, respectively. Under low-pressure and clean atmospheric conditions, it is essential to keep the instrument leakage-free to avoid interference from other sources, e.g., cabin-based emissions. Each aMCPC is therefore subjected to a vacuum leak check while powered off and free of butanol. It is then verified to be completely leak-proof at a vacuum level of 0.067 Pa. In the following, we specifically examine the effects of coincidence, low ambient pressure, and particle diameter on the sampling efficiency of the instrument.

#### 2.1.3 aMCPC Coincidence Correction




In environments with high aerosol concentrations, such as aircraft exhaust plumes, coincidence effects may occur in the laser beam, leading to a non-linear counting behavior of the instrument. Coincidence describes the event when two or more particles coexist simultaneously in the detector's sample volume. The electrical signals produced by light scattering on these particles are inseparable and detected as one. Under ambient aerosol background conditions, this is rarely the case. However, the nearfield turboprop plume concentrations are generally at the order of  $10^4 \, \mathrm{cm}^{-3}$ , which is 1 to 2 orders of magnitude above the background aerosol concentration, and occasionally exceed particle concentrations of  $10^5 \, \mathrm{cm}^{-3}$  as shown in Fig. 6. In this concentration regime, the data must be corrected for coincidence effects. The correction curve in Fig. 5a was obtained experimentally in the laboratory.

The sample aerosol was produced from a miniCAST soot generator with a single mode size distribution around 38 nm particle diameter. The sample aerosol was drawn from a mixing chamber with the same inlet line length as for the aircraft by the aMCPC and a Faraday Cup Electrometer (FCE, GRIMM 5705) (Keck et al., 2009), as a reference instrument. The FCE works on the principle of collecting charged particles in a conductive cup, generating a current proportional to the particle flow, which is not affected by coincidence, low pressure, or particle size. It also requires unipolar charged particles, which is achieved by firstly using a soft X-ray charger to change the balance of irregularly charged particles to a known distribution of  $\pm 0$ V and secondly, a differential mobility analyzer (DMA), which selects unipolar particles according to their electric mobility diameter. A particle diameter of 45 nm was chosen. To account for multiple charged particles in the FCE, additional measurements at 65 and 82 nm diameter were performed, and the raw FCE concentrations were corrected following the procedures detailed in ISO norm ISO 27891:2015(E). By comparing the concentration of the aMCPC simultaneously against the FCE, we find that particle emissions of  $10\,000\,\mathrm{cm^{-3}}$  produce a low coincidence of around 3%, however, increasing to 22% at  $50\,000\,\mathrm{cm^{-3}}$ , and 80% at  $100~000~\rm cm^{-3}$ .

Figure 5a shows the actual concentration  $(N_{\rm FCE})$  from the reference FCE versus the measured aMCPC concentration 190  $(N_{\rm aMCPC})$ . To support the measured relationship with a theoretical consideration, three different models were tested. In Eq. 1, Collins et al. (2013) provide the theoretical solution for the coincidence, assuming a Poisson process. Other commonly used functions to approximate the coincidence are Eq. 2 from Zhang and Liu (1991) or Eq. 3 from Hermann and Wiedensohler (2001); Takegawa and Sakurai (2011) which prove valid only for concentrations below 50 000 cm<sup>-3</sup>.

$$Model 1: N_{FCE} = -\frac{1}{(\tau_d \cdot Q)} \cdot W_0(-N_{aMCPC} \cdot Q \cdot \tau_d)$$
 (1)

$$Model 2: N_{FCE} = \frac{N_{aMCPC}}{\exp(-N_{aMCPC} \cdot Q \cdot \tau_d)}$$
 (2)  
 $Model 3: N_{FCE} = \frac{N_{aMCPC}}{1 - N_{aMCPC} \cdot Q \cdot \tau_d}$  , (3)

$$Model 3: N_{FCE} = \frac{N_{aMCPC}}{1 - N_{aMCPC} \cdot Q \cdot \tau_d} ,$$
 (3)

where the dead time  $\tau_d$  is the time corresponding to the pulse width in the detector, Q the sample flow through the aMCPC, and  $W_0$  the principle branch of the Lambert W function. The dead time  $\tau_d$  is obtained from the fit by model 1 in Fig. 5a. The coincidence effect for particle concentrations greater than  $110~000~\rm cm^{-3}$  shows a steep asymptotic behavior, thus the signal is saturated and errors are above 100%. Therefore, concentrations exceeding  $110~000~\rm cm^{-3}$  cannot be assessed without a dilution system. In our chase sequences, the raw particle concentrations never exceeded  $23~000~\rm cm^{-3}$ , thus the coincidence correction to the concentration was always less than 8% of the data.

# 2.1.4 Low-Pressure Dependent Counting Efficiency of the aMCPC





Our airborne measurements were conducted at FL340 and FL250 corresponding to inlet pressures of 259 and 382 hPa. At lower atmospheric pressure, the partial pressure of the aMCPC's working fluid decreases, resulting in less efficient condensation on aerosol particles. Hence, the low-pressure counting characteristics of the aMCPC must be determined in the lab to apply the correction to the in-flight measurements (Noone and Hansson, 1990; Hermann and Wiedensohler, 2001). In a laboratory experiment, we reduced the pressure gradually while injecting a defined and constant amount of particles into the sampling volume. Silver was evaporated in a furnace at 1100 °C, and particle sizes of 55 nm were selected by a DMA. Using the FCE as a reference instrument, the counting efficiency at inlet pressures between 940 and 180 hPa was inferred. Figure 5b shows the aMCPC counting efficiency (the ratio of aMCPC and FCE number concentration) over the inlet pressure. The counting efficiency starts to decrease at pressures below 400 hPa. Using a Markov-Chain Monte Carlo (MCMC) method (Foreman-Mackey et al., 2013) and  $\eta(p) = a_1 - b_1 e^{-c_1 p}$ , with parameters  $a_1$ ,  $b_1$ , and  $c_1$  as fitting function, we can characterize the low-pressure behavior of the aMCPC and correct the data collected in flight with this fit function. At 380 hPa (FL250), the counting efficiency is at 96%, while it is at 80% for the lowest ambient pressure (260 hPa, FL340) encountered during the measurements.

# 2.1.5 Diameter Dependent Counting Efficiency of the aMCPC

Condensation particle counters have a lower size detection limit due to a lower limit of particle activation at a given butanol pressure. At this so-called cut-off diameter, the efficiency of butanol condensation for activation of the particles is so low, due to the Kelvin effect, that the particles are either not activated or do not grow large enough to be detectable by the measurement of the scattered light. Again, we use an FCE as a reference instrument for the size-resolved detection efficiency of the aMCPC. Particle sizes from 6 to 60 nm were selected using a DMA. In Fig. 5c, the counting efficiency is plotted against the particle geometric diameter. As before, the MCMC method was used to fit the counting efficiency depending on the diameter ( $\eta(D) = a_2 - b_2 e^{-c_2 D}$ , with fit parameters  $a_2$ ,  $b_2$ , and  $c_2$ ). The experiment was conducted at ground-level pressure and two lower pressures relevant to the targeted flight altitudes. The aMCPC concentrations were corrected for the lower pressures using the correction curve shown in Fig. 5b. For this instrument, the  $d_{50}$  diameter - defined as the particle diameter at which the counting efficiency reaches 50% - is 7.5 nm, 8.5 nm, and 8.7 nm at 944 hPa, 411 hPa, and 250 hPa, respectively. If a large nucleation mode is present, ultrafine volatile liquid particles (with diameters 

Figure 5. a) Assessment of coincidence of the aMCPC (section 2.1.3). The FCE vs. aMCPC number concentration is shown. The fits provided by Eq. 1, 2, and 3 show three commonly used coincidence correction models, where it is evident that model 1 from Collins et al. (2013) describes the data best and is therefore used for the correction. b) The counting efficiency of the aMCPC over the ambient pressure. The counting efficiency was determined by the ratio of the number concentration of the aMCPC and an FCE as a reference. With decreasing pressure, the counting efficiency drops, described in section 2.1.4. c) aMCPC counting efficiency as a function of particle diameter for three different pressures (section 2.1.5). The curves show a decrease in counting efficiency at around 15 nm, and the decrease is stronger for low-pressure conditions. d) Combined effect of aMCPC counting efficiency and the transmission efficiency of the inlet system over particle diameters (in the lower size range) for three different pressure levels (section 2.1.6). The reduced efficiency is due to particle deposition on the inlet tubes by diffusion in the laminar flow. Stainless steel tubing, heated inlet lines, and a high bypass flow are used to minimize losses. e) Combined effect of mSEMS and aMCPC counting efficiency and the transmission efficiency of the inlet system over particle diameter. f) Thermodenuder evaporation efficiency of volatile particles dependent on the particle size at three different pressure levels (section 2.1.9).

## 2.1.6 Diffusion Particle Loss of the Inlet Line








Particle losses due to diffusion and deposition on the tubing walls result in the depletion of aerosols within a specific size range. This effect is well known, particularly at low pressures (Baron and Willeke, 2001; Fuchs, 1975). These losses are more relevant for ground-based emission measurements, where the inlet lines and therefore the residence time of the sample gases are significantly longer than for airborne measurements (Schripp et al., 2022). We here assess the effect of the specific sampling line used during the campaign on particle loss at different pressures and aerosol sizes.

From the isokinetic inlet nozzle (UAV inlet, Brechtel Brechtel (2024), Hayward, CA, USA), at the top of the mast, to the instrumentation inside the A-Box, the aerosol passes through a stainless steel tube of approximately 3.5 m length with an inner diameter of 4.6 mm. To avoid significant losses of small particles on the tube walls, the mast sampling line was heated. In addition, a bypass flow of 5.2 lpm generated with a critical orifice and an additional pump inside the A-Box, was used to reduce the residence time in the inlet. This minimized the diffusion losses while maintaining laminar flow.

Figure 5d shows a laboratory measurement with an MCMC fit of the inlet's particle loss and its comparison with the theoretically calculated diffusion losses according to Baron and Willeke (2001), taking into account the tubing length, diameter, and curvature for three different pressure levels between ground and maximum altitude pressure. Both the data and the model show the combined effect of line losses and aMCPC cut-off as described in 2.1.5. The data is naturally affected by the aMCPC's cut-off, and the model is adjusted accordingly. Data were sampled at 914, 400, and 250 hPa to match flight pressures between 380 and 260 hPa and to compare with the ground pressure transmission. The data have been corrected for the low-pressure counting efficiency as described in section 2.1.4. The  $d_{50}$  diameters from the combined effects result in 10, 14, and 16 nm for 914, 400, and 250 hPa, respectively.

A correction of the campaign aMCPC data was applied by making use of the size distribution information provided by the mSEMS (see section 2.1.7).

## 2.1.7 Miniature Scanning Electrical Mobility Sizer (mSEMS)

The mSEMS (miniature Scanning Electrical Mobility Sizer, Brechtel Model 9404) of size  $(0.18 \,\mathrm{m}\,\mathrm{x}\,0.13 \,\mathrm{m}\,\mathrm{x}\,0.10 \,\mathrm{m})$  includes a miniature DMA column that selects particles depending on their electrical mobility. The sample air enters the outer of two concentric cylinders, which function as outer and inner electrodes, respectively. A clean sheath airflow  $(3.0 \,\mathrm{lpm})$  separates the particles from the inner electrode. The charged particles are attracted toward the inner cylinder wall by a voltage ranging from 0 to 3000 V, causing them to migrate through the sheath flow. Depending on the voltage, sheath flow, and charge, particles with a specific diameter enter the downstream sample outlet at the inner electrode (Wang and Flagan, 1990). The mSEMS is able to detect a particle diameter size range from 5 to 375 nm at a minimum scan time of 5 s and a particle concentration range from  $1 \,\mathrm{to}\,10^7 \,\mathrm{cm}^{-3}$ . For our purposes, we use the up and down scanning mode, in which the voltage continuously changes between the lowest and highest values in the size range from 5 to 350 nm with a 30 bin setting, and a constant sheath flow at 3 lpm. In order to do rapid scanning size distribution measurements, the mSEMS is operated with the fast  $(0.18 \,\mathrm{s})$  responding aMCPC.

A soft X-ray charger (XRC-05 by HCTM CO., LTD) is used upstream as a neutralizer. It changes the irregularly charged particles into a bipolar charge equilibrium ( $\pm$ 0 V). The result is a defined bipolar charge distribution.

In the current experiment, we operate the mSEMS with rapid scans over 30 bins with a bin time of 0.5 s of the full size range in order to provide fast and highly resolved size distributions over the plume mode and background. It is reasonable to change the size range for pure exhaust measurements to the Aitken Mode (10 nm to 100 nm) to increase the scan speed. If a nucleation mode is expected, e.g., oil or sulfate particles, a scan should cover the smallest diameters. On the other hand, for sampling atmospheric background only, the bin scan time can be increased to several seconds, allowing for a wider range, including the Accumulation mode, to be scanned.

# 2.1.8 Diameter Dependent Counting Efficiency of the mSEMS

The counting efficiency of the mSEMS for small particles was tested in a laboratory experiment from 6 to 63 nm and is shown in Fig. 5e. The mSEMS uses an aMCPC for particle detection and therefore has the same counting efficiency constraints at small particle sizes as described in section 2.1.5. Additional transmission losses arise from particle diffusion within the X-ray neutralizer and the classifier column, which are described in section 2.1.6. Furthermore, for particles smaller than 10 nm, the probability of acquiring even a single elementary charge in the neutralizer is exceedingly low, thereby limiting the fraction of particles available for classification (Reischl et al., 1996). The fitted curve can be used to correct the data for instrument losses. However, the above-described effects lead to an increase in uncertainties in the size range below 15 nm, which propagate into the derived size distribution.

## 2.1.9 Thermodenuder Evaporation Efficiency





The thermodenuder is a device to discriminate and count particles with a solid core from liquid particles with a defined vapor pressure. The current custom-built version evaporates volatile particles using a 58 cm long and 230 °C heated flow line with a 12 mm inner diameter, according to the principle described in Burtscher et al. (2001). The larger tube diameter reduces the flow velocity, resulting in greater evaporation efficiency. This way, volatile particles can be evaporated, and the remaining nvPM emission can be counted. The evaporation efficiency of the thermodenuder, i.e., the ratio of volatile particles introduced into the thermodenuder to the remaining particles measured, was investigated under laboratory conditions. Ammonium sulfate particles were used as the volatile aerosol and selected by diameter between 10 and 250 nm using a DMA. As in the previous setup, an FCE served as a reference instrument, and both instruments were sampled from an aerosol mixing chamber with identical lengths of inlet tubing to reduce the effects of particle loss due to diffusion losses.

The efficiency to evaporate the volatile aerosol is then determined by Eq. 4 and shown in Fig. 5f for three different pressures.

Thermod. Efficiency = 
$$1 - \frac{N_{\text{Thermod.,aMCPC}}}{N_{\text{FCE}}}$$
 (4)

All volatile particles smaller than 50 nm were completely evaporated by the thermodenuder with a high degree of confidence. If the droplets become too large, the thermodenuder will be unable to evaporate all the particles, as there is insufficient time

and power for evaporation, resulting in a reduction in evaporation efficiency. At 100 nm, the evaporation efficiency is 89.5% for 914 hPa, 94.0% for 400 hPa, and 96.3% for 247 hPa. This is sufficient if the size range of volatile particles is mainly expected in the nucleation mode and the size distribution of soot measurements in engine exhaust conditions peaks around a diameter of 30 nm (Beyersdorf et al., 2015; Moore et al., 2017; Schripp et al., 2018). In combination with the tPM measurement, it is possible to derive information on the number concentration of volatile particles. This can be of particular interest for contrail formation on nucleation mode particles in the low-soot regime (Kärcher and Yu, 2009), or potentially on oil particles, e.g., in the case of hydrogen combustion (Ponsonby et al., 2024; Bier et al., 2024).

# 2.1.10 Optical Particle Counter (OPC)






To detect larger aerosol particles, the A-Box contains an Optical Particle Counter (OPC, SkyOPC model 1.129, Grimm Aerosol, Ainring, Germany) that detects the intensity of light from a 655 nm diode laser scattered by individual aerosol particles. The instrument is operated in "high mode" to detect particles between 0.25 and 2.5  $\mu$ m in 16 channels at 1 Hz. The OPC is also connected to the A-Box pump and has a fixed volume flow of 1.2 (l min<sup>-1</sup>) regulated by a critical orifice. The instrument is calibrated for sizing with NIST-traceable PSL spheres with a refractive index of  $n_r = 1.585$  at 655 nm following the procedure outlined in Walser et al. (2017). The flow is calibrated using a Gilian Gilibrator 2 bubble flow meter (Sensidyne Inc., Clearwater, FL, U.S.A.). Further sources of uncertainty are described in Walser et al. (2017) and stem from the optical sizing method (Mie scattering variability, refractive index assumptions) and counting statistics.

# 310 2.1.11 Summary of Particle Loss Effects

In summary, the uncertainties of the aerosol measurements result from the combined influence of coincidence effects in the aMCPC, pressure- and size-dependent counting efficiencies, diffusion losses in the inlet system, and the charging efficiency of the mSEMS neutralizer. While these processes were quantified in laboratory experiments (Fig. 5), their cumulative impact defines the overall accuracy of the in-flight aerosol observations. For particle diameters above approximately 20 nm, the effective transmission and detection efficiencies remain high (>80% for the aMCPC and >60% for the mSEMS), such that the contributions to the uncertainty are mainly linked to the correction for low-pressure operation (up to 25%) and secondary to the correction for coincidence (< 8%). At diameters below 10-15 nm, the detection efficiency decreases, but this decrease can be corrected for with a size-resolved lab-based measurement. However, the uncertainties increase due to low counting statistics, which come from higher diffusional losses and limitations in charging. The estimated overall uncertainty of the corrected aerosol number concentrations results in the main uncertainty of the determined EI, which consequently leads to an uncertainty of the EI of 18 to 26%. (see section 4.2.1).

# 2.2 CO<sub>2</sub> measurement with Licor 7000

As part of the A-Box, a high frequency ( $\sim 5$  Hz) non-dispersive infrared gas analyzer (LI-COR (b) shown in Fig. 4a) was used to detect  $\mathrm{CO}_2$  mixing ratios. The high sampling frequency enables the capture of the small-scale variability in the turbulent

plume. The Licor-7000 consists of two measuring chambers for the detection of CO<sub>2</sub>: chamber A is permanently supplied 325 with a reference gas (dry synthetic air); chamber B receives ambient air from the inlet at the top of the aircraft-mounted mast. Normally, the instrument is operated with dry synthetic air. During this campaign, nitrogen with ultra-high purity was used instead of dry synthetic air due to limitations in the gas supply. To obtain the absolute mole fractions of CO<sub>2</sub> in dry air, the difference in the absorption of infrared radiation passing through the two cells is calculated (LI-COR, b) and corrected 330 for dilution effects in the post-processing (LI-COR, a). The instrument is modified specifically for aircraft deployment, as the instrument was originally designed for ground measurements. A metal bellows vacuum pump (model MB-602), together with a downstream pressure regulator (LFE), keeps the inlet pressure for the instrument to around  $\sim 1060 \, \text{hPa}$ . The accuracy of the measured CO<sub>2</sub> mixing ratios is approximately 3.4 ppm. This includes the reproducibility of the calibration standards (1.3 ppm), the precision (0.2 ppm), and the uncertainty of the water vapor measurement and therefore the dilution correction 335 (1.9 ppm). Further, the instrument response drifts with instrument temperature (2.5 ppm per maximum instrument temperature change of 8 °C) and flight duration (0.2 ppm per maximum flight duration of 2.5 h). These long-term drifts are accounted for by measuring the reference gas at the ground. For this purpose, two gas sample cylinders (Swagelok type HDF4-1000) were mounted on the instrument assembly. They are filled with synthetic air and a CO<sub>2</sub> reference gas of known concentration, respectively. Software-controlled valves and the respective gas can regulate both gas flows and can be used for drift correction and in-flight calibration. However, long-term drifts are less critical for the measurement of short plume intersections where 340 enhancements above the background are relevant rather than the absolute CO<sub>2</sub> mixing ratios in the background.

Here, the  $\mathrm{CO}_2$  measurements are used to account for dilution in the aircraft wake and to relate the emission species to their relative position in the exhaust plume. This enables the comparison of emission data at different dilution stages in a single plume. In addition, the dilution-corrected emissions expressed in particles per kilogram of fuel burned, known as emission indices, can be used as a metric to compare different engine types and settings (fuel flow, combustion temperatures, thrust, etc.) to assess, e.g., the aerosol particle reduction potential.

# 2.3 $NO_x$ measurement with SIOUX

345

The SIOUX (StratospherIc Observation Unit for nitrogen oXides) instrument (Fig. 4b) is located in the hull of the aircraft and used for  $NO_x$  (=  $NO + NO_2$ ) measurements. To accommodate the 180 kg instrument, the Egrett's airframe was modified. The backward-facing gas inlet is located at the left underwing pod where the CAPS instrument is mounted. The core of the SIOUX instrumentation is a chemiluminescence detector (CLD 790 SR). CLD is a well-established technique for measuring reactive nitrogen species, which are catalytically converted to NO (Bollinger et al., 1983; Fahey et al., 1985) and subsequently detected by chemiluminescence (Ridley and Howlett, 1974; Drummond et al., 1985). Several types of CLD detectors and converters have been used for atmospheric background measurements in the upper troposphere and lower stratosphere aboard the DLR research aircraft Falcon and HALO (Ziereis et al., 2000; Voigt et al., 2005, 2007, 2008; Stratmann et al., 2016; Ziereis et al., 2022), in the upper stratosphere aboard the Russian high-altitude aircraft Geophysica (Schmitt, 2003; Heland et al., 2003). On the Falcon, it has also been used to detect exhaust plumes from aircraft and ships (Schulte and Schlager, 1996; Schlager et al., 1997; Roiger et al., 2015). Aboard the Egrett, the two-channel CLD is capable of measuring NO and simultaneously  $NO_x$  by

converting  $NO_2$  using a blue light converter (Droplet Measurement Technologies, Aircraft BLC). The time resolution of the instrument is  $\sim$ 1 Hz with a detection limit of 110 pptv (1 pptv = 1 pico mol mol^-1) for NO and 130 pptv for  $NO_x$ . For this campaign, the instrument was operated at pressures below 500 hPa, and the conversion efficiency at pressure levels chasing the turboprop aircraft is better than  $\sim$ 90%. Due to difficulties with the gas supply, the SIOUX instrument was calibrated only in the laboratory and not in the field during the mission flights. The uncertainty of the NO ( $NO_2$ ) mixing rations ranges from 20% (80%) at atmospheric background levels to 3% (5%) at the highest detected mixing ratios of  $\sim$ 80 ppb ( $\sim$ 60 ppb) in the sampled aircraft exhaust. The uncertainty is estimated based on CLD-specific parameters of the two channels, see e.g. Stratmann (2013): instrument sensitivity (9790  $\pm$  190/9820  $\pm$ 400 counts ppb $^{-1}$ ), efficiency of the  $NO_2$  converter (90  $\pm$ 5% at 220 hPa), instrument interferences due to desorption processes and dark current (44  $\pm$  107/1021  $\pm$  1308 counts), statistical uncertainty of the count rates 0.02 -0.2 ppb for NO (0.01 -0.15 ppb for  $NO_2$ ), uncertainty in the calibration standard ( $\sim$ 30 ppb), uncertainty in the percentage of NO molecules that do not react with ozone (0.4%), and the uncertainty in the associated instrumental background (300 - 4600 / 1500 - 3500 counts).

# 2.4 Water vapor measurement with WARAN and CR2

Dedicated water vapor measurements are provided by two instruments: The WAter vapoR ANalyzer (WARAN) is a closed-path laser hygrometer, based on the commercial WVSS-II system by SpectraSensors Inc. It derives the concentration of water vapor in the sample flow by using the absorption of the 1.37  $\mu$ m line from an indium–gallium–arsenide (InGaAs) tunable diode laser (TDL) in a closed measurement cell. Mixing ratios between 50 and 40 000 ppm (1 ppm = 1  $\mu$ mol mol<sup>-1</sup>) can be detected with a precision of 5% or 50 ppm, whichever is greater (Voigt et al., 2017; Marsing et al., 2023). With a sampling frequency of 0.3 to 0.4 Hz, it is a relatively fast hygrometer in view of the precision and compact size of the instrument.

A second measurement is done via the CR-2 cryogenic frost point hygrometer from Buck Research Instruments, LLC (Heller et al., 2017) which applies the dew point mirror detection principle. The range of measurable mixing ratios is 1 to 20 000 ppm at a reporting frequency of 0.3 Hz. However, it must be noted that the equilibration time of the frost point measurement at high tropospheric altitudes and low dew points is on the order of tens of seconds. Between 10 and 500 ppm, the precision is 9 to 12%.

Both instruments have been compared by Kaufmann et al. (2014, 2018) and are regularly calibrated in the laboratory against an MBW 373-LX reference dew point mirror. Also for both, custom 1/4" stainless steel inlet lines were fitted for optimal transport of water vapor from the respective backward-facing inlets to the instruments. The WARAN inlet is situated next to the  $NO_x$  inlet, while the CR-2 inlet is placed next to the  $CO_2$  inlet on the top of the mast. This strategy was chosen to provide highly accurate background humidity sampling alongside  $CO_2$  and aerosol background measurements, for accurate relative humidity values. Fast in-plume  $H_2O$  mixing ratios are provided by the WARAN instrument, along with ice particle measurements from the CAPS and  $NO_x$  measurements from the SIOUX. Both water vapor instruments are pumped by membrane pumps of type NMP830KPDC-B4 HP (KNF Micro AG) with volume flows between 3 and 51 min<sup>-1</sup>.

# 2.4.1 Meteorological Parameters

395

400

Static air temperature, (particle) airspeed, and pressure were measured by the CAPS (Cloud and Aerosol Spectrometer Probe) as well as by temperature and pressure sensors on the Cheyenne and Egrett. The temperature sensor is a thermistor (model AD590) with an accuracy of 0.5 K down to a minimal temperature of about 215 K. Prior to this campaign, the temperature and pressure measurements of the CAPS instrument were compared to Falcon onboard sensors (Mallaun et al., 2015) during flight, to account for biases as well as uncertainties. The temperature and pressure were accurate within 1 K and 10 mbar for the speed envelope of the Egrett.

Further, the corresponding meteorological measurements were compared with forecast data from the European Center for Medium-Range Weather Forecasts (ECMWF) and found to be in good agreement with the instrument and model data in most cases, within the limits of detection, atmospheric variability, and the limits of interpolation of the model data onto the flight paths. Only the model temperature was found to be occasionally between 2 and 4 K lower than the measurement for some flight legs. In addition, the reading of the temperature sensor on the chase aircraft was frequently reported by the pilots and agreed within  $\pm$  1 K with the PT100 temperature sensor values of the CAPS probe. This implies a generally high confidence in the meteorological parameters provided here.

#### 405 3 Data evaluation methods of exhaust measurements

The most commonly used metric to quantify aircraft emissions is the emission index (EI). It relates the amount of a species (in terms of number or mass) emitted to the mass of burned fuel. It is derived from in situ measurement data of an emitted substance with known emission characteristics and the simultaneously measured concentration of the particle or trace gas. Measurement of inert tracers with a known amount of emitted gas, such as  $CO_2$ , enables the comparison of other emission products at different dilution stages in a single plume. In addition, the dilution-corrected emission indices, expressed in particles per kilogram of burned fuel, can be used as a metric to compare different engine types and settings (fuel flow, combustion temperatures, thrust, etc.) to assess, for example, the aerosol particle reduction potential.

# 3.1 The Aerosol Emission Index: $EI_{nvPM}$ and $EI_{tPM}$

We determine the emission index for non-volatile particulate matter (soot) and total particulate matter using  $CO_2$  as a dilution tracer. As the  $CO_2$  mixing ratio  $r_{CO_2}$  and the aerosol number concentration  $N_{nvPM}$  and  $N_{tPM}$  are measured from the same inlet position, the signals of these quantities are strongly correlated (see Fig. 7). To account for different sampling frequencies and response times of the instruments, the emission index is determined for each plume encounter by integrating the plume signal over its time span.

$$EI_{x} = \frac{\int_{plume} \Delta N_{x} dt}{\int_{plume} \Delta r_{CO_{2}} dt} \frac{V_{m}}{M_{CO_{2}}} EI_{CO_{2}}, \quad x \in (nvPM, tPM),$$
(5)

where  $\Delta N_x$  denotes the particle number concentration at standard conditions ( $T=273.15\,\mathrm{K}$ ,  $P=1013.25\,\mathrm{hPa}$ ) corrected for coincidence and low-pressure behavior (section 2.1.2), and subtracted by the background concentration. Likewise,  $r_{\Delta\mathrm{CO}_2}$  is the background-subtracted  $\mathrm{CO}_2$  mixing ratio.  $M_{\mathrm{CO}_2}$  is the molar mass for  $\mathrm{CO}_2$ ,  $V_m=22,4\,\mathrm{lmol}^{-1}$  the molar volume at standard conditions and  $\mathrm{EI}_{\mathrm{CO}_2}\approx3160\,\mathrm{g\,kg}^{-1}$  the  $\mathrm{CO}_2$  emission index for Jet A-1 (Moore et al., 2017; Rohkamp et al., 2023). In the high soot regime,  $\mathrm{EI}_{\mathrm{nvPM}}$  and  $\mathrm{EI}_{\mathrm{tPM}}$  are typically in the range of  $10^{14}$  to  $10^{15}\,\mathrm{kg}^{-1}$  for jet engines (Moore et al., 2017; Dischl et al., 2024).

# 3.2 The $NO_x$ Emission Index: $EI_{NO_x}$

The  $NO_x$  emission index ( $EI_{NO_x}$ ) is defined in mass units of  $NO_2$ , i.e. the sum of NO and  $NO_2$  in the plume is considered as if all NO was in the form of  $NO_2$  (ICAO (2008); Voigt et al. (2012); ICAO (2023))

Normally,  $EI_{NO_x}$  is related to the chemically inert dilution tracer  $CO_2$  (Schulte et al., 1997). Here, an approach is provided to derive  $EI_{NO_x}$  using water vapor as the quasi-inert dilution tracer in non-contrail-forming conditions. The inlet positions of the  $NO_x$  and  $H_2O$  measurements are co-located at the landing gear. In the near-field plume, this leads to a better correlation of  $NO_x$  to  $H_2O$  than to  $CO_2$ , of which the inlet is located at the mast. This analysis can only be achieved under non-contrail-forming conditions in near-field plume measurements, as inside contrails and clouds, condensation makes water vapor non-conservative.

 $\mathrm{EI_{NO_x}}$  is determined for each plume encounter by integrating the plume signal over its time span.

$$\operatorname{EI}_{\mathrm{NO}_{\mathbf{x}}} = \frac{\int_{plume} \Delta r_{\mathrm{NO}_{\mathbf{x}}} \, \mathrm{d}t}{\int_{plume} \Delta r_{\mathrm{H}_{2}\mathrm{O}} \, \mathrm{d}t} \, \frac{M_{\mathrm{NO}_{2}}}{M_{\mathrm{H}_{2}\mathrm{O}}} \, \operatorname{EI}_{\mathrm{H}_{2}\mathrm{O}} \quad , \tag{6}$$

where the  $\Delta r$  again indicates the enhancement above background mole fractions.  $EI_{H_2O}$  is the fuel-specific emission index for  $H_2O$  (1250 g kg<sup>-1</sup>) (Schumann, 1996), and  $M_{NO_2}$  and  $M_{H_2O}$  are the molecular masses of  $NO_2$  and  $H_2O$ .

For comparison, we use  $\mathrm{EI}_{\mathrm{NO_x}}$  based on  $\mathrm{r}_{\mathrm{CO_2}}$  in a more homogeneously mixed, well-diluted plume at a 1200m distance.

# 4 Results

In this section, we present the emission indices of tPM, nvPM, and  $NO_x$ , calculated as described in section 3. Further, the aerosol size distribution is analyzed in the plume and in the ambient air, and an  $EI_{tPM}$ -size distribution is derived.

# 4.1 Measurement Sequences of nvPM, tPM, CO<sub>2</sub>, H<sub>2</sub>O, and NO

The chase flights were conducted in close formation, with aircraft distances ranging from 100 m to 1200 m. To highlight the strong gradients between in-plume and ambient background aerosol concentrations, Fig. 6 shows the vertical profile of  $\text{nvPM}_{D_p>10\text{nm}}$ ,  $\text{tPM}_{D_p>10\text{nm}}$ , and the  $\text{CO}_2$  mixing ratio for Flight No. 5 on 13 April 2022. The turboprop chase sequences took place at FL250 (7.6 km, 382 hPa), FL330 (10.0 km, 265/272 hPa), and FL340 (10.4 km, 259 hPa). Background aerosol concentrations vary by roughly 1 order of magnitude around  $5 \times 10^2 \, \text{cm}^{-3}$  at the flight levels where the chase sequences were performed. The aerosol concentration during plume intersections is 2 to 3 orders of magnitude higher than the background, with

Figure 6. Pressure profile of  $nvPM_{D_p>10nm}$ ,  $tPM_{D_p>10nm}$  number concentration and  $CO_2$  mixing ratio for Flight No. 5 on April 13. The enhanced  $CO_2$  measurements indicate aircraft emissions at FL250 and FL330/FL340 corresponding to aerosol number concentration exceeding the ambient concentrations by about 2 orders of magnitude.

maximum number concentrations of 23  $000\,\mathrm{cm^{-3}}$ .  $\mathrm{CO_2}$  mixing ratios of up to 106 ppm above the background of  $\sim$  421 ppm have been measured during the plume measurements.




Figure 7 shows an example time series of the near-field emissions of  $nvPM_{D_p>10nm}$ ,  $tPM_{D_p>10nm}$ ,  $H_2O$ ,  $CO_2$ , and NO at FL250. The aerosol particle emissions correlate strongly with the  $CO_2$  tracer measurements, which are obtained using fast-responding instruments and sampling both quantities from the same inlet location (mast) (see section 2). The  $H_2O$  and  $NO_x$  measurements correlate strongly due to the same inlet position (landing gear), but weakly with the  $CO_2$  and aerosol signals, as the plume dimensions in the near-field are relatively small compared to the extension of the Egrett inlet positions (see section 1.1). The position of the inlets relative to the plume center impacts the correlation of the trace gas measurements. Therefore, we observe sections where either the plume was sampled with the mast or with the landing gear inlet, hence simultaneous measurements of background and in-plume conditions are possible, which is intended by design for the measurement of ambient aerosol activation for hydrogen combustion.

To provide an overview of the flights, Table 1 shows the respective flight levels, in-flight meteorological, and engine parameters during the turboprop chase (No. 3, 5 out of a series of flights) on April 11 and 13. Chase Flight No. 3 contains one

Figure 7. Example of a timeline of turboprop near-field emission measurements at FL250. The upper two graphs show the timelines of the aerosol number concentration of  $nvPM_{D_p>10nm}$  and  $tPM_{D_p>10nm}$  and  $CO_2$  mixing ratio above the background  $\Delta r_{CO_2}$ , both with inlets at the mast. The lower two panels show the timelines of  $H_2O$  and NO mixing ratios above the background measured with inlets at the landing gear.

measurement sequence at FL330, whereas in Flight No. 5, the turboprop emissions were recorded during the entire flight at FL340, FL330, and FL250. The calculated EIs are also provided in Fig. 8 and their uncertainty is discussed in the following.

# 4.2 In-flight Emission Indices of nvPM and tPM

We derive aerosol emission indices for 69 plume encounters, resulting in a total of 30 min of data. Figure 8a shows the median  $EI_{nvPM}$  and  $EI_{tPM}$  of all definite plume crossings with its 25% and 75% percentiles for the three flight levels.  $EI_{tPM}$  range from (9.6 to 16.2)  $\times 10^{14} \, kg^{-1}$  and  $EI_{nvPM}$  from (8.1 to 12.4)  $\times 10^{14} \, kg^{-1}$ , listed in Table 1. The emission indices are on the

| Flight Level                                                                 | 340          | 330          | 250           |
|------------------------------------------------------------------------------|--------------|--------------|---------------|
| Flight No.                                                                   | F5.1         | F5.2 / F3    | F5.3          |
| $P_{ m static}$ / hPa                                                        | 259          | 265 / 272    | 382           |
| SAT / °C                                                                     | -43          | -43 / -48    | -29           |
| RHw / %                                                                      | 10           | 11/15        | 4             |
| EGT / °C                                                                     | 415          | 420 / 400    | 350-360       |
| Fuel Flow / kg/h                                                             | 90.7         | 90.7 / 90.7  | 97.5          |
| PAS m/s                                                                      | 108          | 108 / 104.5  | 93.4          |
| KIAS / knts                                                                  | 135          | 135 / 135    | 135           |
| ${ m EI_{NO_x,median}}$ / ${ m gkg^{-1}}$                                    | -            | 7.7          | 7.3           |
| ${ m EI_{NO_x,[25/75]}}$ / ${ m gkg}^{-1}$                                   | -            | [6.8 / 8.3]  | [6.9 / 8.4]   |
| ${\rm EI_{nvPM,median}}  /  10^{14} \; kg^{-1} \; ({\rm D_p} > 10 {\rm nm})$ | 9.3          | 8.1          | 12.4          |
| ${\rm EI_{nvPM,[25/75]}}/10^{14}~{\rm kg^{-1}}$                              | [8.7 / 10.4] | [7.8 / 9.3]  | [11.6 / 14.3] |
| $\rm EI_{\rm tPM,median}$ / $10^{14}~kg^{-1}~(D_{\rm p}>10{\rm nm})$         | 11.0         | 9.6          | 16.2          |
| $\rm EI_{tPM,[25/75]} / 10^{14}  kg^{-1}$                                    | [9.8 / 13.2] | [9.1 / 11.7] | [15.0 / 18.3] |

Table 1. In-flight meteorological parameters as well as engine parameters during the measurement sequences sorted by flight level (FL). The upper part of the table lists in-flight recorded static pressure ( $P_{\rm static}$ ), static air temperature (SAT), relative humidity with respect to water (RHw), exhaust gas temperature (EGT), fuel flow, particle air speed (PAS), and knots indicated air speed (KIAS). FL330 was probed on two different days, hence it shows two sets of meteorological parameters. The lower part shows the calculated median for  $EI_{\rm tPM}$ ,  $EI_{\rm nvPM}$ , and  $EI_{\rm NO_x}$  with their 25 and 75 percentiles.

order of  $10^{15}$  particles per kg fuel burned. A comparison of  $EI_{\rm tPM}$  and  $EI_{\rm nvPM}$  shows that the largest amount of engine-emitted particles (tPM) consists of soot (nvPM), i.e., 85%/85%/77% at FL340/FL330/FL250, which is in agreement with recent turbofan measurements from Dischl et al. (2024). However, the plume contains on average between 15% and 23% vPM (tPM-nvPM) corresponding to  $EI_{\rm vPM}$  of  $1.70\times10^{14}\,{\rm kg^{-1}}$  and  $3.8\times10^{14}\,{\rm kg^{-1}}$  at FL340 and FL250, respectively.



A Wilcoxon–Mann–Whitney test was used with a significance threshold of 5%. Using this statistical method, from FL340 to FL330  $\rm EI_{nvPM}$  and  $\rm EI_{tPM}$  decrease by 15% however, with p values larger than 5%, indicating no statistically significant difference. With an increase in fuel flow of only 7% from FL340 to FL250,  $\rm EI_{nvPM}$  and  $\rm EI_{tPM}$  increase significantly by 53% and 69%, respectively. However, taking into account the measurement uncertainties and the uncertainty on the statistical representativeness of the samples taken, these changes have to be investigated in more detail in future studies.

Measurements of engine emissions at cruise altitude for comparison are sparse. In particular, missing information on particle emissions from in-flight or ground measurements of turboprops to compare with only allows comparison with turbofan and turbojet engine emissions. Moore et al. (2017) and Dischl et al. (2024) show that for the large turbofan engines at cruise conditions,  $\mathrm{EI}_{\mathrm{nvPM}}$  for conventional petroleum-based jet fuels is on the order of  $10^{14}-10^{15}\,\mathrm{kg}^{-1}$ . At distances corresponding

to about one minute of contrail age, larger particle emission indices of up to  $5 \times 10^{15} \, \mathrm{kg^{-1}}$  were observed (Voigt et al., 2021; Bräuer et al., 2021a). Our measurements are in agreement within the order of magnitude of previous jet engine emission measurements. This is expected due to similar combustion processes of the engines. This agreement translates back into a consistent set of aerosol and trace gas measurements, while simultaneously adding to the current database of in-flight emission data. A more systematic measurement, conducted under various ambient and engine conditions both in flight and on the ground, would be required to demonstrate the representativeness and comprehensiveness of the measurements. In particular, our data, in conjunction with ground-based emission data from the LTO cycle, which are proprietary to the engine manufacturer, would be valuable for validating scaling methods from ground to altitude, as demonstrated in Schulte et al. (1997); Dischl et al. (2024); Harlass et al. (2024).

# 4.2.1 Uncertainty of $EI_{nvPM}$ and $EI_{tPM}$





To account for inlet line losses for small diameters (see sections 2.1.5 and 2.1.6), we use the in-plume size distribution measured with the mSEMS (see section 4.4). Since the main mode of the aerosol distribution is in the soot size range and losses by diffusion or detection are relevant for smaller particles, we expect a systematic underestimation of the number concentration of 10%, and its correction (ll) with an error of  $\Delta_{\rm err} ll = 5\%$ . The error of the low-pressure counting efficiency correction  $\Delta_{\rm err} lp$  was estimated to be 15% for the respective flight levels. The error of the CO<sub>2</sub> mixing ratio is  $\Delta_{\rm err} r_{\rm CO_2} = 3.4\,{\rm ppm}$  and the variation in the background mixing ratio is  $\Delta_{\rm err} bg_{\rm CO_2} = 1\,{\rm ppm}$ . The error of EI<sub>CO<sub>2</sub></sub> that results from the accuracy of the hydrogen-to-carbon molar ratio of the fuel is relatively small and neglected here. Thus, the relative error of the emission index is derived as follows:

$$\Delta EI_{x} = \sqrt{\left(\frac{\partial EI_{x}}{\partial N_{x}} \Delta_{err} N_{x}\right)^{2} + \left(\frac{\partial EI_{x}}{\partial lp} \Delta lp\right)^{2} + \left(\frac{\partial EI_{x}}{\partial ll} \Delta ll\right)^{2} + \left(\frac{\partial \Delta EI_{x}}{\partial CO_{2}}\right)^{2} \left(\Delta_{err} CO_{2}^{2} + \Delta_{err} BG_{CO_{2}}^{2}\right)}$$
(7)

This leads to an uncertainty in  $EI_x$  of 18 to 26%, which results mainly from the uncertainty of the correction of sampling efficiency at low pressures.

# 4.3 In-flight Emission Indices of NO<sub>x</sub>

In contrast to  $EI_{\rm nvPM}$ , we derive  $EI_{\rm NO_x}$  with the measurement of water vapor described in section 3.2. The strong correlation between  $NO_x$  and  $H_2O$  with both inlets at the same position (unlike  $CO_2$ ) results in a better statistical representation. The evaluation is based on 10 plume encounters and 11 min of measurement time. We determine  $EI_{\rm NO_x}$  for FL 250 and FL330, while for FL340 the sampling time for  $NO_x$  and  $H_2O$  was too short to calculate an emission index. The medians with their 25% and 75% percentiles are shown in Fig. 8 and listed in Table 1. The median  $EI_{\rm NO_x}$  is 7.3 and 7.7 g kg<sup>-1</sup> for FL330 and FL250, respectively.

Since water vapor is a non-conservative quantity due to condensation on aerosols within the plume, the method's accuracy may be reduced, particularly at low temperatures, high relative humidities, and high surface area densities. To account for this, we derive  $EI_{\rm H_2O}$  experimentally from the in-plume measurements of  $r_{\rm H_2O}$  and  $r_{\rm CO_2}$  from plume intersects at equal distances. The

experimentally derived  $EI_{H_2O}$  is  $1116\,g\,kg^{-1}$  with an uncertainty of  $\pm 15\,\%$  at FL250. Despite the 10% lower value compared to the theoretical value of  $1250\,g\,kg^{-1}$ , both  $EI_{H_2O}$  agree within the uncertainties of the measurement. Therefore, no measurable change in engine water vapor due to condensation on ambient or plume aerosol is observed.

Based on these considerations, we derive an uncertainty of  $\mathrm{EI}_{\mathrm{NO}_{\mathrm{x}}}$  of 15%, governed by the accuracy of the  $\mathrm{NO}_{\mathrm{x}}$  instrument, described in section 2.3 and 2.4.

An additional estimation of  $EI_{NO_x}$  using  $CO_2$  as a tracer was performed for a short measurement sequence during a single plume encounter at the largest distance of 1200 m. At this distance, a quasi-homogeneous plume concentration is assumed, reducing the impact of different inlet positions. We derived an emission index of  $5.3\,\mathrm{g\,kg^{-1}}$  with a large uncertainty of 30%. This value is 27 to 31% lower than the medians derived using  $H_2O$  as dilution tracer, yet confirming the  $EI_{NO_x}$  values within the uncertainty of the measurements. In summary, these values provide an upper estimate of the  $NO_x$  emission index, as a reduction of water vapor due to condensation would lead to lower  $r_{H_2O}$  and therefore larger  $EI_{NO_x}$  (Eq. 6). To set these low  $EI_{NO_x}$  values into perspective, we compare our measurements to previous ground and in-flight measurements. For several modern turbofan engines,  $EI_{NO_x}$  values between 8.4 to 19.7 g kg<sup>-1</sup> for FL between 328 and 350 have been reported (Schulte et al., 1997; Jurkat et al., 2011; Harlass et al., 2024). Turbofan engines tend to produce more  $NO_x$  than turboprop engines due to the temperature-dependent nature of  $NO_x$  formation, i.e., higher combustion temperatures and pressures in turbofan engines. Laboratory-based measurements of a small turboshaft engine (313 kW maximum shaft power) reported by Rohkamp et al. (2023) revealed  $EI_{NO_x}$  values ranging from 4.06 to 5.33 g/kg at 30% to 100% of maximum shaft power.

For the turboprop investigated here, we find similar emission indices for tPM and nvPM compared to large turbofan engine emission measurements. However, its  $\mathrm{EI}_{\mathrm{NO}_{\mathrm{x}}}$  values are lower than those of turbofan engines but align more with ground-based turboshaft emission measurements. Therefore, our measurements confirm that turboprop engines have  $\mathrm{EI}_{\mathrm{NO}_{\mathrm{x}}}$  values at the lower end of turbofan engines and agree with the current knowledge of combustion processes and reported emission indices.

# 4.4 Aerosol Particle and Emission Index Size Distribution





In this section, we provide size distributions of  $EI_{tPM}$  and geometric mean diameters of the in-flight aerosol measurements of the mSEMS behind the Cheyenne. The data are taken at FL330 during Flight No. 3 with measured static atmospheric temperature and pressure of  $47.92 \pm 0.24$  °C and  $272.27 \pm 0.71$  hPa, respectively. Due to power issues, the mSEMS was not operational during Flight No. 5. Figure 9a shows the combined mSEMS and the OPC data in a log-log plot, covering a total range of 5 nm to  $2.5 \,\mu\text{m}$ . In contrast to the tPM distribution in ambient air (blue), the tPM in-plume size distribution (red) shows a mode in the soot-coagulating regime around 30 nm. The OPC is set to a recording time of 1 s, while the mSEMS average scan time was 17 s. This results in large differences between individual scans, leading to the shown variability.

From the particle size distribution recorded by the mSEMS shown in Fig. 9, the distribution of  $\rm EI_{tPM}$  can be deduced. The background-corrected and STP-converted distribution scans from the instrument are used in Eq. 5 with the integrated  $\rm CO_2$  mixing ratio over the time of a scan. High variability of aerosol concentrations in the plume leads to a high variability of the

Figure 8. Median aerosol and  $NO_x$  emission indices at different fuel flows and flight levels with 25th and 75th percentiles. The upper plot shows the  $EI_{tPM}$  and  $EI_{vnPM}$  in particles per kg of burned fuel. The lower plot shows  $EI_{NO_x}$  in g kg<sup>-1</sup> of burned fuel.

derived EI and thus a larger standard deviation. A log-normal distribution (Eq. 8) was fitted to the data:


$$545 \quad \frac{dEI_{tPM}}{d\log D} = \frac{EI_{tPM}}{\sqrt{2\pi}\log(\sigma_g)} \exp\left(-\frac{(\log D - \log D_g)^2}{2(\log \sigma_g)^2}\right) \quad , \tag{8}$$

where  $\frac{dEI_{tPM}}{d\log D}$  is the bin normalized  $EI_{tPM}$  of tPM, D the particle diameter, and  $D_g$  and  $\sigma_g$  the geometric mean diameter and geometric standard deviation, respectively. A fit of  $EI_{tPM}$  data results in  $D_g = 27.5 \pm 2.0$  nm. Thus, the main mode of the size distribution presented here is predominantly in the soot size range, with only a small fraction of smaller particles being detected at this early plume age. This may be due to either reduced sampling efficiency of the small particles, coagulation of particles, or a combination of both.

Aerosol size distribution measurements from in-flight exhaust sampling have only been reported twice (Schröder et al., 2000; Moore et al., 2017). The latest  $\rm EI_{nvPM}$  distributions reported by Moore et al. (2017) provide a mean geometric diameter of  $27.8 \pm 0.3\,\mathrm{nm}$  for tPM and  $32.5 \pm 0.4\,\mathrm{nm}$  for nvPM and are thus comparable to our measurements.

Figure 9. a) Size distributions of tPM at FL330 at 272 hPa and 225 K. The red line shows the mean of the distributions measured in the plume (including the ambient concentrations) at a distance of  $105 - 319 \,\mathrm{m}$   $(1.0 - 3.2 \,\mathrm{s})$  while the blue line represents just the ambient aerosol distribution at the same flight level over a flight segment of 29 km. The shading represents the standard deviation of the measurement variability and the propagated uncertainties from the correction described in section 2.1.8. The bars above show the range of the mSEMS, which covers the vast majority of the particle sizes, and the detection range of the OPC, with a strong decrease in the number of particles with diameters above 200 nm. In contrast to the broad mean distribution of the ambient air, the mean in-plume distribution shows a clear mode around 30 nm. b) Mean tPM emission index size distribution  $dEI_{tPM}/dlog_{10}D$  (with standard deviation) of plume segments of 90 s total measurement. Calculated from the data shown on the left using Eq. 8. Large variations occur due to the variability in the in-plume distributions. However, the mean of the distribution is well described by a log-normal distribution. A fit using the Eq. 8 gives a geometric mean diameter of  $D_g = 27.5 \pm 2.0 \,\mathrm{nm}$ .

# 5 Conclusions and Outlook

A Grob Egrett was equipped with a new set of instruments for  $CO_2$ ,  $NO_x$ , water vapor, and aerosol measurements that operated autonomously during flight at altitudes between 7.6 and 10.4 km (FL250 and FL340). They were successfully tested for inplume measurements of a turboprop Garrett/Honeywell TPE 331-14 engine. For the first time, the results provide insight into the in-flight emission characteristics of a small turboprop aircraft. In particular, we quantify the aerosol particle emissions co-located with  $CO_2$  emissions, and  $NO_x$  emissions co-located with water vapor emissions to determine in-flight emission indices. We conclude that in non-contrail forming conditions, water vapor can be used as a conservative tracer to derive  $EI_{NO_x}$ , which is a requirement for non-hydrocarbon fuels such as direct  $H_2$  combustion. Analysis of the emission index for both nvPM and tPM demonstrated that the aerosol emissions predominantly consist of soot particles, although a notable fraction of volatile particles (up to 23% of tPM) is also emitted, comparable with previous jet emission measurements. The behavior of the nvPM and tPM number concentration over plume age is briefly shown and discussed in a supplement to this paper. Additionally, the ratio of nvPM to tPM (given as the ratio of  $EI_{nvPM}$  to  $EI_{tPM}$ ) is shown over the plume age. While both concentrations

dilute with plume age, the data are too sparse to make a well-founded statement about the ratio, where plume aging or particle modification through aggregation, growth, or scavenging could be assessed.

Although lacking a dedicated measurement program, we provide  $\mathrm{EI}_{\mathrm{nvPM}}$  and  $\mathrm{EI}_{\mathrm{tPM}}$  with varying engine and fuel flow settings. In particular,  $\mathrm{EI}_{\mathrm{NO}_{\times}}$  showed a very low value of  $\sim$ 7.5 g/kg compared to the typical emissions indices of higher-thrust jet engines.

Additionally, the aerosol size distributions were measured in the exhaust plume and atmospheric background. Due to large gradients from sampling in the near-field with the mSEMS, the size-resolved emission index distributions vary substantially. Nevertheless, significant differences from the ambient aerosol distributions were observed, revealing a mode within the soot accumulation regime following a log-normal distribution with geometric mean and standard deviation at  $D_{\rm g}=27.5\pm2.0\,{\rm nm}$ . Since this geometric diameter falls within the range of jet engine soot emissions measured in flight, it likewise enhances the confidence of our measurements. If the expected size distribution is known, the measurements of future plumes can be optimized regarding scan times. Further, longer plume intersections would increase the mSEMS accuracy. The need for emission measurements of new technologies, either from demonstrators or new engines entering service, is greater than ever, as emission measurements in the jet regime (up to 5 s of plume age past emission (Kärcher et al., 2015)) provide the basis for assessing the climate impact of these technologies. Future measurements of non- ${\rm CO}_2$  effects of turboprops, such as contrail formation and  ${\rm NO}_x$  emissions, should target larger passenger aircraft at relevant cruise altitudes with a wide range of engine conditions to provide a better reference and benchmark in terms of size and weight for future hydrogen-propelled aircraft.

*Data availability.* Data are archived at the HALO database at https://halo-db.pa.op.dlr.de/mission/144 (HALO, 2024). Free primary data access begins on 20 April 2026. Until then, download access and access to further data will be given upon request.

Author contributions. TJW, CR, and JC planned and coordinated the project. AV and RV own the aircraft and operated the aircraft during the measurement. DS, CH, and PS planned the construction and instrumentation of the A-Box. GN performed the characterization of the A-Box aerosol instruments and data evaluation and wrote the paper. AM contributed to the description and evaluation of water vapor measurements. TH, SB, and MP contributed to the description and evaluation of nitrogen oxide measurements. AM, CH, CV, DS, GN, PS, TJW, and TH performed the campaign measurements and data evaluation. VH and EDLTC contributed to instrument calibrations. All authors contributed to the paper.

Competing interests. CR and JC are employed by Airbus Operations. RV and AV are employed by AV Experts LLC. All the other authors declare that they have no conflict of interest.

Acknowledgements. Flight hours were financed by Airbus. AM and TH were funded by Airbus. TJW, DS, CH, GN, and PS acknowledge funding by the DLR H2-Contrail project. Special thanks go to the AV Experts engineers and pilots who provided support throughout the campaign. We appreciate the fruitful comments of Honeywell representatives on the data analysis.

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
