# Peer review of "In-flight emission measurements with an autonomous payload behind a turboprop aircraft"

_EGUsphere, 2025_

## Author Comment (AC1)

**Answer to referee 1 comments for "In-flight emission measurements with an autonomous payload behind a turboprop aircraft"**

*We would like to thank the reviewer for the suggestions, which will improve the manuscript substantially. We will address the comments point by point below in italic font.*

**General Comments**

Summary: The impact is well-motivated, showing the importance of studying short range flights with turboprop aircraft. Significance of the impact, difficulty of the measurement, and lack of comparable datasets highly motivate the scientific significance. The method is sound and calibrations and subsequent analyses are thorough, including variance and error propagation. Some explanation as to why the aerosol instruments need to be pressurized would be preferable, since I was confused by the explanation of sampling stability. I am suspicious of the particle sizing and concentration accuracy at the small sizes ~10nm, due to the cut size of the particle counters (exacerbated by diffusion losses) leading to large correction factors, thus the mode size may be slightly overestimated; however, results compare well to previous airborne measurements with similar instrumentation from Moore et al., 2017. The author did a very good job characterizing instrument response to sample and environmental factors, and uncertainty. There are a couple of typos.

*We thank the reviewer for this positive assessment and will address the specific comments below.*

**Specific Comments**

L15-17: Are the size distributions presented from the total or nonvolatile aerosol? If total, suggest removing "soot" and generalizing as jet engine emissions, since the soot implies non-volatile particulates.
*Agreed, the size distribution of the total aerosol is now explicitly specified. The observation that the discovered mode falls within the range of previously measured jet engine soot suggests that soot is the primary contributor to this distribution.*

L25-26: Incomplete sentence. Suggest combining with previous sentence.
*Thank you, we have clarified this sentence.*

L97: L228 is the first time bringing up isokinetic, but it would be helpful to mention at the introduction of the aerosol inlet.
*Agreed, we added this.*

L122-123: Can you expand on what "ensuring stable sampling conditions" means? Why would the instruments need to be in a pressurized vessel? It introduces a higher deltaP and increases the potential for dilution/leak into the sample.
*One key reason is that the Licor instrument for $CO_2$ measurements needs to be operated at cabin pressures. Additionally, the MCPCs, SEMS, and their associated computer systems were not originally designed for operation at such high altitudes and low pressures. The ceiling altitude of the measurements was unclear at the time of system design. To avoid the risk of electrical issues such as arcing, we chose not to operate them under these conditions. We discussed the possibility of testing the instruments in a cloud chamber. However, based on many hours of laboratory testing, where we operated with low pressure in the sample line,*

*we chose to -run the instruments in a pressurized environment to enhance the stability of the measurements during the flights. You are correct that, in general, a higher deltaP increases the potential for leaks. For this reason, we repeatedly tested the system for leaks under high deltaP conditions during the campaign (by reducing the inlet pressure and sampling from a filter). For the tubing within the instruments, which were mostly Tygon tubing, we expected a higher impermeability.*

L206-208 & F5b: The curve fit deviates from the measured counting efficiency right around your critical operating environment. The operating environment is barely captured in your data points. Higher resolution in the region that you measure (more points between 375-250 hPa) would result in more precise correction.

*Thank you for this comment. We agree that additional measurement points between 250 and 350 hPa would have better constrained the fit in this range. We also acknowledge that the model may not capture the MCPC behavior perfectly in this range, however, considering all data points, the fit converges well. The figures below show the subset of the Markov Chain Monte Carlo (MCMC) results with the model median and the 1 σ credible interval. While the model median lies outside the 1 σ range of the experimental data at 300 hPa, it remains within the credible interval from the MCMC posterior. More measurements would probably not alter the median fit curve but rather constrain the 16th and 86th percentile. We therefore retained the model together with its associated uncertainty, which is propagated in our error analysis to account for this local deviation.*

[Figure]

L223: Is the 90-degree tube bend sufficiently large enough radius to be negligible for inertial impaction of large particles? Valuable to mention if negligible here when describing other loss mechanisms.

*Following the theoretical particle loss approach described by Baron & Willeke (2001), we used P. Baron's particle loss calculator to estimate losses of larger particles due to inertial effects. The calculation included all sample lines, including their diameters and curvatures, as well as the flow rates of sample air going through the sampling lines. Results indicate that particle losses become significant for diameters exceeding 1 μm. Although the OPC's size range includes such particles, we intentionally excluded them from our analysis because OPC concentrations drop to near zero for particles of D > 400 nm.*

L259: Typo. soot soot.
*Thank you, we fixed that.*

L267: Why the range in sheath flow? Is the range from intentional changes, e.g., compensating for pressure to achieve the same size range in a scan, or fluctuation due to environment? What is the corresponding sample flow?

*The range given here refers solely to the instrument's capability. The transfer function (the probability that an aerosol particle entering the SEMS will exit through the detector outlet) depends on the sheath flow, and a higher sheath flow narrows the classification window, thereby improving resolution. To achieve high resolution in short time intervals, we selected a sheath flow of 3 L min⁻¹ for our measurements. This flow is maintained by an internal pump and was verified to be constant in the recorded data, although those data are not shown here. The sample flow was approximately 0.36 L min⁻¹.*

L270: Typo on mSEMS. "Is able to operate at 5 s scan time", does that mean you did operate a 5 s scan? What was the lag time from the sample out of the DMA to the aMCPC? For a 5-second scan, the smearing may be significant. You don't mention operation scan times until L509, and it's worth mentioning how it was operated in section 2.1.8. 17s scan while in a highly variable plume seems too slow for samples shown in F7 without a large lag chamber. Were scans averaged to suppress the noise, and if so, how many scans are used for averaging?

*- Thank you, the typo is fixed. 5s refers to the instrument's shortest possible scan time, although this setting was not used during these flights.*
*-The delay time between the SEMS and MCPC is 0.9 s; both up- and down-scans were evaluated under laboratory conditions to determine this delay time.*
*- As noted in the comment, we could only use scans clearly identified as in-plume, resulting in a total effective measurement time of just over 1 min. The conditions were sufficiently stable during these scans with clear $CO_2$ enhancements above the background. We acknowledge that this represents extremely sparse data. These data are from the first flight, in which the turboprop chase was the shorter of two planned experiments. For this initial run, we planned longer SEMS scan times to account for the variable background conditions. We acknowledge that a shorter scan time would have been advantageous in this case and had planned to adjust the settings accordingly for subsequent flights.*
*Unfortunately, we only had a shorter chase sequence during the first flight, and during the main chase flight, the SEMS failed to start due to a power issue.*

L289-290: Is it supposed to read "sizing" instead of "size"? Why are instrument sizing and flow calibration major sources of uncertainty? Are you talking about the physical size due to unknown refractive indices?
*Thank you! Yes, it should read "sizing". We have changed the wording.*
*The OPC flow is controlled with a critical orifice, which fixes the volume flow. It is not a major source of uncertainty, and we change that. We included the reference to Walser et al. (2017), who further describe the uncertainties in the optical sizing method. We don't examine the OPC data with the same detail as for other instruments, as its importance is secondary to our evaluation.*

F9, F5c, F5d: The combined effects of the aMCPC size cut, counting efficiency with pressure, and diffusion losses, hurts the confidence in the size distributions below 20 nm. I expect the entire left-side falling edge of the curve in Figure 9 in plume would have increasing error bars associated with it, which may provide context/caution in interpreting the mode size from the fit in F9b. The aMCPC may not be the best choice for engine emission characterization since its cut size is near to the exhaust particle size range. There may be a significant number of sub-10 nm particles missing from the tPM when calculating the EI. Perhaps the EI should be specified as the EI_tPM>10nm at the top of the document. This may be less of a concern for

this generation of engine, but consider ultra fine CPCs when testing future generation engines that combust more efficiently that they may have a smaller mode size where the aMCPC will completely misinform/bias the peak.

*We agree to specify the measurements as >10 nm, since it falls slightly below the D50 cutoff under our sampling conditions. Thank you for this comment. We have now included the efficiency losses for small sizes in Figure 9. We also included the efficiency calibration of the SEMS + MCPC setup into the instrument analysis in Fig. 5e and section 2.1.8. This illustrates the combined effect of SEMS and MCPC on the particle cut-off.  The calibration uncertainties propagate into the corrected data and show now a realistic estimation for the error at <10 nm particle diameters. While the correction did not change the general appearance of the distribution, it did shift the mode from Dg = 34.7 ± 1.9 nm to Dg = 27.5 ± 2.0 nm.*

---

## Author Comment (AC2)

**Answer to referee 2 comments for "In-flight emission measurements with an autonomous payload behind a turboprop aircraft"**

*We would like to thank the reviewer for the suggestions to improve the manuscript. Below, you will find our responses to their comments. The reviewer's comments are written in normal font, and our answers are in italics.*

**General Comments**

The paper describes a new in-flight measurement capability for measuring a range gaseous and particle emissions from aircraft engines. The work describes the first application of the method measuring behind a turboprop. Data from turboprops, both in-flight and on the ground, are rare and this provides an extremely useful dataset to the community. The application of two inlets, allowing for simultaneous in-plume and background, is a really nice feature.

The paper is very well written, with particular attention paid to the detail. The uncertainty in the measurements is well treated. I only have a few minor comments, which are listed below.

*We thank the reviewer for this positive assessment and will address the specific comments below.*

**Specific Comments**

In the abstract, define cruise altitude (Line 5)
*The definition of cruise altitude is given two sentences later. We moved it now to the first time it was mentioned.*

Line 14: do you mean lower than expected NOx based on predicted? Please clarify (based on previous, based on ground-based predictions?). Could it be caused by using H2O instead of CO2 for the calculation (see below)?
*As there is no other publicly available TP emission data, the NOx emission index is low compared to known measurements of typical emission indices of modern higher-thrust jet engines, as later referenced in Harlass et al. 2024. It is, however, not an unrealistic value considering that the engine is an older turboprop engine with lower pressure ratios than modern engines.*
*It is unlikely that the use of $H_2O$ as a tracer caused this, as the relative humidity during the measurement was low enough (Table 1) to consider water vapor an inert tracer. Moreover, an underestimation of water vapor would cause the calculation in Eq. 6 to yield a higher emission index for $NO_x$.*

Line 14, tPM, nvPM or vPM size distributions?
*Our setup measured the tPM size distribution. Thank you for pointing it out. We added it to the description here.*

Line 29, delete comma after Both
*We removed this.*

Line 41, do you have a reference to a roadmap or similar to the potential for these new technologies?
*We agree that this is needed. We included an FAA and an IATA roadmap as a reference to this statement.*
*https://www.faa.gov/aircraft/air_cert/step/disciplines/propulsion_systems/hydrogen-fueled_aircraft_roadmap ("Even at projected 3kW/kg by 2035 fuel cells may be best suited for aircraft carrying fewer than 75 passengers and short-haul flights").*
*https://www.iata.org/contentassets/8d19e716636a47c184e7221c77563c93/aircraft-technology-net-zero-roadmap.pdf*
*("ZeroAvia plans to deliver a 9-19 seater hydrogen fuel cell powered aircraft in 2025, and a 40-80 seater by 2027.")*

Figure 2 is referenced before figure 1 (line 87).
*True, we swapped figure 1 and figure 2.*

There needs to be a bit more detail on the operational details of the Egrett (operating altitudes, range, science speed range)
*The information is added: "The chase aircraft, a Grob G 520 Egrett (Fig. 1), is a high-altitude and long-endurance turboprop aircraft with a certified maximum operating altitude of 13,716 m (45,000 ft) (Grob aircraft SE), a maximum airspeed of 463 km/h (250 kn), and a range of 4260 km (2,300 Nmi) with an endurance of 8.0 hours dependent on payload and weather (NASA Airborne Science Program). Operated by AV Experts LLC, the Egrett was suited to test the instruments and to perform measurements in the near-field exhaust plume (100 - 1200 m) and background atmosphere."*

Figure 5b. The model is outside of the error bars of the data exactly at pressures where most of the data is collected. There either needs to be more points added to constrain the fit better or this needs to be incorporated into the error analysis.
*Thank you for this comment. We agree that additional measurement points between 250 and 350 hPa would have better constrained the fit in this range. We also acknowledge that the model may not capture the Mixing Condensation Particle Counter (MCPC) behavior perfectly here. The figures below show the subset of the MCMC results with the model median and the 1 σ credible interval. While the model median lies slightly outside the 1 σ range of the experimental data at 300 hPa, it remains within the credible interval from the MCMC posterior. Furthermore, the model considers all data points and their associated uncertainties. Adding more measurements there would not substantially change the fit curve but certainly reduce the uncertainties. We therefore retained the model together with its associated uncertainty, which is propagated in our error analysis to account for this local*

*deviation.*

[Figure]

Figure 5d, It is not clear what the data is. The Y axis is labelled as aMCPC, but the figure legend is inlet system losses. Can this be explained more clearly, and which curve or curves are used in the loss correction section?

*The Y-label can indeed be a bit confusing. It represents the inlet line losses in addition to the MCPC's cut-off losses. It therefore represents the combined effect of particle-size-related losses, as described in Section 2.1.6. It is now described more specific in the figure caption.*

Line 267 – why is there a range of sheath flows? Have you verified this is not changing during a scan? Has this been incorporated into the error analysis (changing the sheath to aerosol ratio changing the resolution of the DMA etc)?

*The range given here refers solely to the instrument's capability. Since the transfer function (the probability that an aerosol particle entering the SEMS will exit through the detector outlet) depends on the sheath flow, and a higher sheath flow narrows the sizing window, thereby improving resolution. We selected a sheath flow of 3 L min$^{-1}$ for our measurements, which was constantly used throughout the measurements. This flow is maintained by an internal pump and was verified to be constant in the recorded data, although those data are not shown here.*

Line 464 – where does the value of 10% undercounting and subsequent correction come from? That needs clarification.

*This is calculated by assuming that the particles follow the unimodal lognormal size distribution that we have discovered in the plume. Applying the losses due to size (aMCPC cut off and inlet line diffusion losses) to this distribution, the difference in the number concentration following from the integrated distribution is ~10%.*

The EINOx using the water vapor is an interesting approach. I can see not having CO2 co-located with the Nox at short distances might be an issue as one inlet may be in the plume and the other not. What I would like to see is the EInumber calculated with CO2 and H2O(CR2) from the box A inlet as these should give the same value and give confidence in the EINOx approach based on the author's claim that the WARAN and CR2 agree well.

*We agree that the approach of calculating aerosol particle EI with water vapor is an interesting method, which we will apply for later missions. However, for this mission it is not possible to calculate the aerosol particle EI with $H_2O$ as the CR2 is a temperature-based measurement system with a resolution similar to that of the WARAN, but with a slow response to changes in water vapor. It primarily captures limited changes in the $H_2O$ mixing ratio typical of ambient conditions, rather than the rapid variations during in-plume measurements (We refered to these measurement capabilities in section 2.4 "However, it*

*must be noted that the equilibration time of the frost point measurement at high tropospheric altitudes and low dew points is on the order of tens of seconds.").* The CR2, however, provides more reliable detection of low ambient water concentrations. The statement of good agreement between WARAN and CR2 refers to the ambient mixing ratio.

Section 2.1.8, 4.4 and figure 9 – the paper goes into great detail on the uncertainties in the measurement system, but I do not see that same detail for the mSEMS or the OPC. For the mSEMS, the extremely low charging probabilities make quantification challenging. 100 particles at 10nm in dN/dlogDp space over a 5 second scan, corrected for charging efficiency, is an incredibly small number of particles getting to the MCPC detector. Is the variability in the data as shown by the red shaded area in figure 9 really larger than the uncertainty associated with the instrument and conditions (short scan time (smearing), low numbers, charging probabilities, possible changing DMA resolution)? Are the smallest sizes in the distribution a true representation of the PSD?
*Thank you very much for this comment.*
*- We have now included the efficiency losses for small sizes in Figure 9. We also included the efficiency calibration of the SEMS + MCPC setup into the instrument analysis in Fig. 5e with a description in section 2.1.8. This illustrates the combined effect of SEMS and MCPC on the particle cut-off. The calibration uncertainties propagate into the corrected data and show now a realistic estimation for the error at <10 nm particle diameters. While the correction did not change the general appearance of the distribution, it did shift the mode from $D_g = 34.7 \pm 1.9$ nm to $D_g = 27.5 \pm 2.0$ nm.*
*- The OPC uncertainties were left out in this scope as they are discussed in great detail in Walser et al., 2017, [https://doi.org/10.5194/amt-10-4341-2017](https://doi.org/10.5194/amt-10-4341-2017), or in the PhD Thesis [https://edoc.ub.uni-muenchen.de/21664/](https://edoc.ub.uni-muenchen.de/21664/)*

Purely out of curiosity, given the relatively simple equations being used, would Monte Carlo simulations be a simpler and more accurate method of calculating the uncertainty rather than the full error equation?
*Thank you for this comment. We agree that a Monte Carlo approach could provide a more general treatment, particularly when error terms are not strictly independent or normally distributed. In our case, we chose the analytical formulation because it makes the contribution of each term to the total uncertainty explicitly visible, allowing the reader to see which factors dominate the error budget.*

---

## Author Comment (AC3)

**Answer to referee 3 comments for "In-flight emission measurements with an autonomous payload behind a turboprop aircraft"**

*We would like to thank the reviewer for the suggestions to improve the manuscript. Below, you will find our responses to their comments. The reviewer's comments are written in normal font, and our answers are in italics.*

**General Comments**

The manuscript describes the instruments comprising an autonomous payload for measurement of CO2, NOx, H2O and particles and their initial deployment on a Grob Egrett aircraft to sample the exhaust emissions from a light turboprop aircraft at cruise altitude. The manuscript is well organized and the descriptions and analysis are clear and fairly comprehensive. The topic is certainly relevant for AMT and I recommend it for publication with only minor comments and suggestions for the authors.

*We thank the reviewer for this positive assessment and will address the specific comments below.*

Minor comments:

L2:   perhaps "on the successful first deployment of"
*Accepted*

L3:   could clarify that you are measuring the exhaust of other aircraft
 *We agree and added this information to the sentence.*

L3:   perhaps "custom-built and commercially"
*We agree and have changed the wording.*

L5:   "temperatures and pressures"; "performed these first"
*We agree and have changed the wording.*

L6:   suggest "a Piper Cheyenne, a twin-turboprop aircraft powered by…"
*We agree and have changed the wording.*

L11:   suggest omitting ", which is adequate"
*We agree and have changed the wording.*

L74:   "non-CO2 effects from aircraft emissions"
*We agree and have changed the wording.*

L75:   "of the instrument payload for a chase aircraft"; size distribution not included here?
*Yes, it is included in the tPM measurements. We revised that sentence to provide a higher level of detail.*

L81:   Could omit paragraph, or be a little more explicit—"Further" seems vague
*We agree and have changed the wording.*

L88:   "near-field exhaust plume"
*We agree and have changed the wording.*

L96:  omit "specifically" and maybe "accommodate"
*We agree and have changed the wording.*

Fig 1 caption:  "Piper Cheyenne (400LS, registration 30 N92EV)"
*Changed. Thank you!*

L130: the heated section evaporates the volatile material, the subsequent cooled section is to lower the temperature prior to introduction into the CPC, right?
*Yes, but also to condense volatile gaseous material on the tubing walls.*

L136: omit the last sentence
*We agree and have changed the wording.*

Fig 3: "puring" ◊ ""purging"; the pumps associated with the aMCPCs are for the saturator flow, not a sheath flow, correct? The mSEMS does have a sheath flow—pump not shown?
*Thank you for pointing this out. There was indeed some incorrect naming. The main pump at the exhaust is responsible for drawing the MCPC sample flows, while the saturator flow pumps only regulate the saturator flow. You are also correct that the sheath flow of the mSEMS is controlled by its own pump. We have updated the schematic accordingly.*

L166: "as is shown in Fig. 6."
*Accepted*

Fig 5 (and subsequent uncertainty discussion): it would be nice to have a panel that shows the combined uncertainty of the various factors that are shown separately and a discussion of the overall magnitude to conclude section 2.1
*Thank you, we included a short summarizing discussion at the end of the section.*

L212: "or do not grow large enough"
*Accepted*

L223: "deposition" would be a better word than "sedimentation", or you could just say "diffusion to the tubing walls"
*Thank you, we changed this!*

L226: why do ground-based measurements necessarily require longer inlet lines and residence times?
*Not necessarily; it has more of a practical and technical reason, as it is difficult or impossible to place instrumentation near the aircraft engine or plume (it is too hot or too turbulent). Both instruments and operators are subject to safety concerns.  On the ground, these instruments are often housed in a measurement container or trailer, rather than being placed directly next to the exhaust. The tubing must bridge that gap. Ground campaigns often compare several instruments or sampling configurations (dilution stages, conditioning, filters). This adds additional tubing length.*

L230: is the heating of the sampling line "to avoid significant losses of small particles on the tube walls" mechanism thermophoresis? Or are you preventing ice build-up? How warm? For what length is there a thermal gradient?
*Yes, the effect is based on thermophoresis. The sampling line was heated equally over the full length; the exact temperature was, however, not recorded. With additional isolation, we aim*

*to achieve temperatures around the freezing point, primarily to prevent water vapor from condensing.*
*Ice buildup is generally captured by an anti-ice installation, which requires more heat at the tip of the inlet where the coldest temperature prevails. We did not account for anti-ice heating and therefore could not fly in supercooled clouds.*

L246: inner diameter?
*Yes, thank you.*

L259: "soot soot"
*Thank you.*

L362: "referred to as "particle" speed because that is the speed CAPS observes particles to travel? Otherwise "True Air Speed" is the more recognized parameter
*The CAPS is technically measuring the true are speed which is referred to as Particle air speed because it is the speed measured at the probe. This is done by a Pitot tube and a pressure sensor. Due to ramp pressure effects at higher speeds, the PAS may be smaller than the TAS. In our case, the PAS is equivalent to the TAS; however, as the TAS measurement of an aircraft has a defined position, there can be slight differences between the measurements.*

L375: "in situ" is not hyphenated
*Thank you.*

L382: "as a dilution"
*Thank you, we changed that.*

L401: "near-field"; "measurements, as inside contrails and clouds condensation makes water vapor non-conservative."
*Thank you.*

L406: Schumann ref in parentheses? Sig figs on molecular seem excessive—actually could omit the number altogether.
*Agreed.*

L414: "vertical profile"
*Thank you.*

L421: "example"; "emissions of"
*Thank you.*

Fig 7: time series of Nnv / Nt would be interesting to see; "near-field" in caption
*We agree that the variability of the ratio of $N_{nvPM}$ /$N_{tPM}$ is an interesting aspect. However, we believe that it cannot be assessed on a 1Hz basis, but rather the ratio of the sum of $N_{nvPM}$ to the sum of $N_{tPM}$ over each plume encounter is a better measure. This is reflected in the ratio of $EI_{nvPM}$/$EI_{tPM}$, which we now discuss in a supplement added to the article.*

L437: "on the order of"
*Thank you.*

L450: Clause including "slightly aged about one-minute-old" is awkward
*Yes, we changed that.*

L461: First sentence is unnecessary
*Thank you*

L465: "low pressure counting"
*Disagree. Without the hyphen, 'low' could be read as modifying' pressure counting' as a whole, which is unclear.*

L466: not sure what is meant by "corresponding"
*It is a bit vague. We changed it to "respective".*

L493: compare to what ground and in-flight measurements? "previous" of …
*Agreed, we added "previous"*

L543: what is "jet-phase"?
*Often referred to as the jet regime. There is no exact definition. It describes the very near-field stage right behind the engine exit, where the hot exhaust jet is dominated by strong turbulence, shear, and rapid mixing with ambient air. In this region, temperatures, pressures, and chemical species are far from ambient, and microphysical processes (soot, ion clusters, sulfur chemistry) are highly dynamic. Kärcher & Yu (2009) define the jet regime as "up to approximately 5 s of plume age past emission".*

L547: "data are archived"
Thank you.

*We thank the reviewer for the careful reading and the many detailed comments, which helped us to improve the manuscript.*